# Have the "mega-journals" reached the limits to growth?

Bo-Christer Björk

Hanken School of Economics, Department of Management and Organisation, Helsinki, Finland

## ABSTRACT

A "mega-journal" is a new type of scientific journal that publishes freely accessible articles, which have been peer reviewed for scientific trustworthiness, but leaves it to the readers to decide which articles are of interest and importance to them. In the wake of the phenomenal success of PLOS ONE, several other publishers have recently started mega-journals. This article presents the evolution of mega-journals since 2010 in terms of article publication rates. The fastest growth seems to have ebbed out at around 35,000 annual articles for the 14 journals combined. Acceptance rates are in the range of 50–70%, and speed of publication is around 3–5 months. Common features in mega-journals are alternative impact metrics, easy reusability of figures and data, post-publication discussions and portable reviews from other journals.

## INTRODUCTION

The scientific scholarly journal emerged in the 17th century and evolved as an institution into its current form in the mid 20th century. A traditional scientific journal appears with a number of regular issues, the format of the articles is more or less standardized depending on the subject field, and the way articles are accepted and edited follows peer review routines which most academics become intimately familiar with during their careers as authors, reviewers and editors. Readers get access to the articles mainly through the subscriptions of their institutions (*Ware & Mabe, 2015*).

Little has seemingly changed since the advent of the web, except for the delivery medium, which nowadays is predominantly electronic, rather than as paper issues. However, as in so many other fields, the web is also acting as a "disruptive innovation," providing opportunities for fundamentally reshaping the processes and business models. In particular, this has provided the platform for the Open Access movement, first started by academics, but later involving start-up publishing companies. There are currently slightly over 10,000 scholarly Open Access (OA) journals indexed in the Directory of Open Access Journals (DOAJ) (http://doaj.org/, 23.4.2015, 10,469 journals).

Due to their electronic-only delivery, Open Access (OA) journals have been able to experiment with many of the "conventions" of scientific publishing. An obvious one is the restriction to publishing articles in a limited number of regular issues. A second one is the

Corresponding author
Bo-Christer Björk,
Bo-Christer.Bjork@hanken.fi

fundamental business model for financing the publishing operations. A third one is the way the peer review works.

With the emergence, around the turn of the millennium, of start-up publishers like BioMedCentral and Public Library of Science (PLOS), the business model of scholarly journal publishing has been reversed. Rather than charging readers, publishers charge the authors in order to fund their work, usually by requiring the payment of article process charges (APCs). The most evident benefit for the authors is the global open accessibility. The share of APC charging journals among full Open Access journals has rapidly increased in recent years (*Laakso & Björk, 2012*), and leading commercial and society publishers are now rapidly founding full OA journals. In addition to pure OA journals, over 8,000 subscription journals nowadays also allow a paid OA option in so-called hybrid journals (*Björk & Solomon, 2014*).

Both issueless publishing and financing operations via APCs have rapidly become the mainstream of open access publishing, but most OA journals still adhere to traditional peer review practices. Articles also look pretty much the same as in traditional subscription based journals. The volume range of scientific journals has traditionally been broad, from small journals in the social sciences and humanities with only a single yearly issue to prestigious biomedical journals appearing as frequently as weekly. In the print format, few journals have exceeded 1,000 research articles per year. Before the Internet, the microeconomics of paper publishing and selling subscriptions made such journals rare exceptions.

Similarly, the scope of journals has varied enormously. Very old publishers such as major scientific societies have tended to publish broad journals congruent with their constituencies (*Journal of the American Medical Society*), and some of the world-leading scientific journals are extremely broad (*Nature*). After the second world war in particular, the leading commercial publishers started creating lot of "niche" journals picking up new emerging research themes (*Electronic Commerce Research and Applications*). In the social sciences and humanities, journals have sometimes been established for extremely narrow purposes (*Nordic Wittgenstein Review*).

The early successful electronic-only OA journals were usually of the narrow variety, often focusing on phenomena relating to the Internet (*Journal of Medical Internet Research*). When new professional OA publishers emerged, they at first strived to launch journals using the same peer review mechanisms and criteria as traditional ones (*PLOS Biology*). In 2006, the OA publisher Public Library of Science launched a new type of journal, which later, due to its success, was been termed a "mega-journal" (the term has also been spelled megajournal or mega journal, in this article mega-journal will be used). *PLOS ONE* incorporates a number of features into its business model which all have been used or at least experimented with before but never in this unique combination. These features have been discussed by a number of authors (*Frank, 2012*; *Binfield, 2013*). The central criterion is a new type of peer review which only checks that the research methods are scientifically trustworthy but does no evaluation of the perceived scientific impact or contribution. Provided that the quality of submissions overall is reasonable this leads

**Table 1  Criteria for a mega-journal used in this study.**

| Criteria/mentioned in sources | Wikipedia | Binfield | Frank |
|---|:---:|:---:|:---:|
| **Primary criteria** | | | |
| Big publishing volume or aiming for it | ✓ | ✓ | ✓ |
| Peer review of scientific soundness only | ✓ | ✓ | ✓ |
| Broad subject area | ✓ | ✓ | |
| Full open access with APC | ✓ | ✓ | ✓ |
| **Secondary criteria** | | | |
| Moderate APC | | | ✓ |
| High-prestige publisher | | | |
| Academic editors | ✓ | ✓ | ✓ |
| Reusable graphics & data | | | |
| Altmetrics | | | |
| Commenting | | | ✓ |
| Portable reviews | | | |
| Rapid publication | ✓ | | ✓ |

to a lower rejection rate than usual. Together with the broad scope, this opens up the opportunity for very big publication volumes.

Inspired by the phenomenal growth of *PLOS ONE*, several reputable publishers have started similar journals in the past 3–4 years, all apparently aiming at annual publication volumes of hundreds if not thousands of articles. Since this is a very recent phenomenon, hardly any research has been done about it. The aims of this study were to:

- Identify currently existing mega-journals
- Estimate the number of articles published longitudinally since 2010
- Collect data about the publication charges
- Collect data about the acceptance rates
- Collect information about the lead times from submission to publication
- Study other aspects of interest.

## METHODS

It is unclear to this author who first coined the phrase mega-journal, and hence first discussed what criteria such a journal should fulfill. The Wikipedia article on the topic starts out by saying that "A mega journal is a peer-reviewed academic open access journal designed to be much larger than a traditional journal by exerting low selectivity among accepted articles" (*Wikipedia, 2015*)[1]. This sentence conveys the rationale for the choice of the term mega, indicating a business plan to publish clearly more articles than scholarly journals usually do. The articles by *Binfield (2013)* and *Frank (2012)* also include explicit lists of criteria. These sources have been taken into consideration in synthesizing the definition below in Table 1, used in the study at hand. The three columns indicate which criteria have been mentioned in the three sources above.

[1] The author wishes to emphazise that he hasn't made any contribution to the Wikipedia article.

Binfield and Frank do not use exactly the same terms as above; both, for instance, talk indirectly about the aim for a big publishing volume in discussing automated workflows, marginal cost for each article below the APC, etc.

The criteria have been grouped into two levels: primary and secondary. In order to qualify as a mega-journal, a journal has to fulfill all the primary criteria. A mega-journal should also fulfill most of the secondary criteria, although they are not mandatory. The APC is in this study considered moderate if it's 1,500 USD or below. This is around the average article processing charge currently paid by academics in Western European countries (*Björk & Solomon, 2014*) and slightly above the price charge by mega-journal market leader *PLOS ONE*. The publisher prestige criterion is important in an indirect way since it is a very important factor in attracting manuscripts early on and in assuring rapid inclusion in the Web of Science. The impact factor then further accelerates the growth of the journal. This criterion thus helps in identifying strong mega-journal candidates before they have reached high publication volumes and even in the announcement stage. Academic editors means that academics, not professional employed staff, act as editors of the individual manuscripts.

Many mega-journals contain value-adding features, which for the time being mostly are lacking in traditional journals (even in the electronic versions). The reusability of graphics and data is facilitated by the use of Creative Commons licenses, which allows readers to reuse materials without explicitly asking for permission, and technically by having options for downloading images in high resolution for reuse in say presentation slides. Public Library of Science has been a pioneer for what is often termed "Altmetrics" (*Priem et al., 2010*). This means that the article contains constantly updated data about citations, bookmarks and tweets. Authors (as well as readers) can get also detailed breakdowns of how the article has been downloaded. The possibility for readers to comment on articles provides a sort of open peer review. Also, authors can use such facilities for responding.

An interesting but somewhat controversial practice that several mega-journals are adopting is the reuse of reviews from journals which have rejected the manuscript in question. The terms 'cascading' and 'portable reviews' have been used to describe this (*Davis, 2010*). Usually the originating journals are more selective journals of the same publisher but in some cases even cross-publisher co-operation has been announced (*Clarke, 2013*).

A recent systematic study of publishing delays in different scientific fields (*Björk & Solomon, 2013*) found average publishing delays of between 9 and 18 months depending of the field of science. In biomedicine, the area of a majority of mega-journal articles, the average was 9.5 months. For OA journals, the average was 6 months. In the context of this study, rapid publication was defined as less than half a year from the original submission.

The identification of mega-journals that fulfill the primary criteria and most of the secondary ones was based on the author's previous knowledge of the OA market, on web searches and on tracking citations to earlier reports about mega-journals. Hence, it was a bit like detective work and there is a risk that some recent journals, which could have fitted the criteria, may have been missed.

**Table 2** The mega-journals studied with basic information about them.

| Journal | APC USD | Publisher | Subject field | Impact factor |
|---|---|---|---|---|
| AIP Advances | 1,350 | American Institute of Physics | Physics | 1.6 |
| Biology Open | 1,350 | The Company of biologists | Biology | WoS |
| BMJ Open | 1,875 | BMJ Publishing Group | Medicine | 2.1 |
| Elementa, Science of the Antropocene | 1,450 | BioOne | Earth sciences | |
| FEBS Open Bio | 1,200 | Elsevier | Molecular sciences | WoS |
| G3 | 1,950 | Genetics Society of America | Genetics | 2.5 |
| IEEE Access | 1,750 | IEEE | Electronics | |
| Journal of Engineering | 1,150 | IET | Engineering | |
| PeerJ | 400 | PeerJ | Biology, medicine | WoS |
| PLOS ONE | 1,350 | PLOS | Science, medicine | 3.5 |
| Royal Society Open Science | 1,600 | Royal Society | All sciences | |
| SAGE Open | 195 | SAGE | Social science | |
| Scientific Reports | 1,495 | Nature Publishing Group | Natural sciences | 5.1 |
| Springer Plus | 1,085 | Springer | All sciences | WoS |

## RESULTS

### List of megajournals

The basic information about the fourteen identified journals is shown in Table 2. There is a clear dominance of biomedicine in terms of topics. Several of the journals already have impact factors and others will get them very soon.

The list of potential journals was longer but journals were dropped due to a number of reasons. One was that some candidate journals don't explicitly have a review process using the soundness-only review criterion. Hindawi's *Scientific World Journal*, which published almost 3,000 articles in 2014, was not included for this reason. *ELife* fullfils some of the other criterions, but is aiming to be highly selective in prejudging the impact of an article. Also, starting journals like *Modern languages Open* (Liverpool University Press), *Open Linguistics* (De Gruyter) and *Science Advances* (AAAS) were not included for the same reason. *Brill Open Humanities* (Brill) and *Heliyon* (Elsevier) would fill the criteria but have not started publishing yet. *Optics Express* has a high publication volume but a very narrow subject field. Many so-called predatory publishers have journals with extremely broad subject areas aiming at high volumes (*Beall, 2013*), but they don't fulfill some of the other criteria on the list, in particular since their peer review practices are often highly deficient.

### Article volumes

The year 2010 was picked as the starting point for the collection of article volumes, because this was the year that competitors for *PLOS ONE* appeared on the market. Two methods were used to determine the yearly article numbers. For journals indexed in Scopus, a targeted search for articles and reviews was made in that index; this yielded quantitative results for most of the journals with big publication volumes. For the year 2014, some of the bigger journals were also checked from their web sites, since the Scopus figures for the

**Table 3 Development of article volumes in mega-journals 2010–2015.** The figures for 2015 are the articles published in the first quarter of the year multiplied by four.

| Journal/articles per year | 2010 | 2011 | 2012 | 2013 | 2014 | 2015 | All years |
|---|---|---|---|---|---|---|---|
| PLOS ONE | 6,913 | 13,833 | 23,441 | 31,882 | 30,054 | 22,120 | 128,243 |
| Scientific Reports | | 211 | 800 | 2,553 | 3,941 | 7,692 | 15,197 |
| BMJ Open | | 109 | 655 | 959 | 1,037 | 1,080 | 3,840 |
| Springer Plus | | | 82 | 692 | 762 | 612 | 2,148 |
| AIP Advances | | 255 | 380 | 396 | 509 | 548 | 2,088 |
| PeerJ | | | | 232 | 471 | 636 | 1,339 |
| G3 | | 65 | 257 | 250 | 272 | 176 | 1,020 |
| SAGE Open | | 44 | 113 | 225 | 217 | 360 | 959 |
| Biology Open | | | 143 | 162 | 137 | 160 | 602 |
| FEBS Open Bio | | 4 | 52 | 78 | 120 | 120 | 374 |
| Royal Society Open Science | | | | | 50 | 196 | 246 |
| IEEE Access | | | | 63 | 106 | 64 | 233 |
| Journal of Engineering | | | | 20 | 102 | 80 | 202 |
| Elementa | | | | 13 | 16 | 28 | 57 |
| All journals | 6,913 | 14,521 | 25,923 | 37,525 | 37,794 | 33,872 | 156,548 |

last year did not appear to be complete at the time of analysis. For the rest of the journals their websites were studied to obtain the article numbers. The figures for 2015 are an estimate based on multiplying the articles published in the first quarter of the year by four. The longitudinal development, is shown in Table 3. The order is descending according to the total number of articles. It should be noted that at present only six journals have volumes above 500 articles per year.

These statistics seem to indicate that the total output of the mega-journals reached a plateau in 2013 after which *PLOS* has started to lose market share, in particular to Scientific Reports.

## Publications charges (APCs)

The APCs of the journals studied varied in the range 195–1,950 USD, with several journals situated close to the 1,350 USD charged by *PLOS ONE*; the average was 1,300 USD. This level can be compared to a global average for OA journals of around 900 USD (*Solomon & Björk, 2012*), charges of 2,500–5,000 USD in top-ranking OA journals and the general APC level of 3000 USD in hybrid OA journals (*Björk, 2012*).

Most of the mega-journals have announced a price and kept it unchanged. A notable exception *is SAGE Open*, which quite dramatically lowered its APC from the initial 695 to 99 USD in 2013. The reason given for this was that an author survey (*Research Information, 2013*) revealed that most authors who are in the social sciences and humanities had to pay the APC out of their own pocket (currently the APC of *Sage Open* is 195 USD).

*PeerJ* offers a membership model in which the author pays a one-time fee and can publish as many articles as he wishes. The fee varies according to how many articles per year are allowed, and varies between 99 for one article per year and 299 USD for an

unlimited number. However, every co-author needs to be a member. The de facto APC in total paid per article is difficult to calculate because it also depends on the number of repeat authors. *Davis (2014)* in an analysis of 600 articles published in the journal found a median number of four authors per article, which would indicate an average per article cost of 400 USD. This would be a probably upper limit to the long run average APC.

## Acceptance rates

The only way to obtain information about the acceptance rates for individual journals is if the publisher in question has made this information available. Acceptance rate data could be found for the following journals:

*PLOS ONE*, 69% (http://www.plosone.org/static/information).

*BMJ Open*, 60% (http://bmjopen.bmj.com/site/about/).

*Scientific Reports*, 55 (http://occamstypewriter.org/trading-knowledge/2012/07/09/megajournals/).

*FEBS Bio Open*, 68%, (http://occamstypewriter.org/trading-knowledge/2012/07/09/megajournals/).

*Biology Open*, 51%, (http://bio.biologists.org/site/about/about_bio.xhtml).

These acceptance rates can be compared to the acceptance rates of scholarly journals in general, Although some top journals have acceptance rates of only 5–10%, *Sugimoto et al. (2013)* in a study of around 5,000 journals in five broad areas found average acceptance rates of between 30% (business) and 46% (health). The corresponding acceptance rates for OA journals in these same fields were between 37% and 57%.

What this means is that risk of rejection in a mega-journal is significantly lower than in the average peer-review journal in its field. Authors who are reasonably competent as researchers probably perceive the rejection risk as very low, and also much less prone to the subjective values and possible biases of the reviewers asked to judge the significance of the results in more selective journals.

## Publishing speed

Some mega-journal publishers have made information about acceptance and publishing public on the journal website, in press releases, editorials, conference presentations etc. *Scientific Reports* for instance announces a median time of 139 days for February 2015 articles (http://www.nature.com/content/srep/statistics/index.html?WT.mc_id=WEB_SciReports_2014_LP). *BMJ Open* reports a median time to first decision of 46 days in 2013, but there is no info of the full publishing delay (http://bmjopen.bmj.com/site/about/). In December 2014, articles published in *PLOS ONE* took 123 days to go from submission to acceptance and 30 days more to final publication (*Davis, 2015*). *Springer Plus* claims a speed to first decision of less than three months, and less than a week from acceptance to publication (http://cofactorscience.com/blog/journal/springerplus). For articles published in the first half year of *PeerJ*, the median time from submission to final acceptance was 51 days (https://peerj.com/blog/post/60259877854/how-fast-is-peerj/).

Based on the figures above, mega-journals seem to have shorter throughput times compared to OA journals in general, not to mention subscription journals. So-called

Table 4 Secondary features in use in the studied mega-journals.

| Journal/features | Moderate APC | High-prestige publisher | Academic editors | Reusable graphics & data | Altmetrics | Commenting | Portable reviews | Rapid publ. |
|---|---|---|---|---|---|---|---|---|
| AIP Advances | ✓ | ✓ | ✓ | ✓ | ✓ | ✓ | ✓ | ? |
| Biology Open | ✓ | ✓ | ✓ | ✓ | ✓ | | ✓ | ? |
| BMJ Open | | ✓ | | ✓ | ✓ | ✓ | | ✓ |
| Elementa | ✓ | ✓ | ✓ | ✓ | ✓ | | | ? |
| FEBS Open Bio | ✓ | ✓ | ✓ | ✓ | ✓ | | | ? |
| G3 | | ✓ | ✓ | ✓ | ✓ | ✓ | ✓ | ? |
| IEEE Access | | ✓ | ✓ | | ✓ | ✓ | | ? |
| Journal of Engineering | | ✓ | ✓ | | ✓ | ✓ | | ? |
| PeerJ | ✓ | | ✓ | ✓ | ✓ | ✓ | ✓ | ✓ |
| PLOS ONE | ✓ | ✓ | ✓ | ✓ | ✓ | ✓ | ✓ | ✓ |
| Royal Society Open Science | | ✓ | ✓ | ✓ | ✓ | | ✓ | ? |
| SAGE Open | ✓ | ✓ | | ✓ | ✓ | ✓ | ✓ | ? |
| Scientific Reports | ✓ | ✓ | ✓ | ✓ | ✓ | | ✓ | ✓ |
| Springer Plus | ✓ | ✓ | | ✓ | ✓ | ✓ | ✓ | ✓ |

predatory OA journals seem to be even quicker than mega-journals (*Shen & Björk, 2015*), but they often lack a proper peer review and lack academic credibility.

## Secondary features

Table 4 shows the inclusion of the secondary features in the studied journals. It is highly likely that all journals fulfill the publishing speed criterion, but for those were no statistics could be found a "?" has been inserted in the table.

*PeerJ* isn't backed by an established publisher, but on the other hand its founders and management team includes individuals with a proven track record in scholarly publishing.

## DISCUSSION

Fourteen mega-journals were identified. In some cases the publishing volumes are still rather small but, given the backing of well-known publishers and the high probability for getting impacts factors in the near future, the growth potential is there. The mega-journal as a type of scholarly journal has emerged rapidly, as demonstrated by the growth in publication volumes reported in Table 3. Publishers have identified a clear market demand for this type of publication outlet and are offering their services at a price level which attracts a lot of submissions. A key competitive factor in launching a mega-journal is if the journal can use the publisher's brand to create a high-quality image from the start, hence names like BMJ Open, Springer Plus, SAGE Open. A mega-journal published by a major publisher can also benefit from a ready IT infrastructure for review and manuscript management.

On the other hand, the first rapid growth seems to have leveled out and competitors of market leader *PLOS ONE* seem to be capturing an increasing share. At the current

level of just under 40,000 annual articles, the output still only represents some 2% of the yearly global output of scholarly articles, which is around 2 million (*Ware & Mabe, 2015*). However, the market share of mega-journals of the APC financed Open Access market is much bigger. Extrapolating from the figure of 136,000 and the growth rate reported in *Laakso & Björk (2012)* as well as from the figure of 120,000 articles by OASPA member publishers (*Redhead, 2014*), a conservative estimate for DOAJ registered OA journals would be 200,000, which would mean that the share of mega-journals would be in the order of 15–20%. The future development of mega-journals could be highly dependant on the growth of APC charging OA journals in general.

Mega-journals have found a place in the scholarly publishing ecosystem where they get a fair share of their manuscripts via rejections from other journals higher up in the ecosystem (usually highly selective journals). In his survey of authors in four mega-journals *Solomon (2014)* found resubmission rates of between 32 and 62%, depending on the journal. The use of portable reviews means that in many such cases the authors lose relatively little time in getting published. One important aspect of mega-journals is how they complement the portfolios of major commercial and society publishers consisting of subscription journals, most of which nowadays offer a hybrid OA choice, and niche and more selective Open Access journals. The mega-journals can use the same technical infrastructures as the other journals, and provide an easy way for the publishers to tap into the growing APC market. The rapid launching of mega-journals by many established publishers indicates that they are seen as commercially viable. However, each major publisher can, credibly have only one mega-journal in its portfolio, which poses a limit to the viable number of journals with strong publisher branding.

Although mega-journals offer excellent economies of scale, the growth could in the longer run be limited by the availability of motivated unpaid reviewers (*Buriak, 2015*). Firstly, the task is less intellectually challenging than in more selective traditional journals, where the perceived contribution needs to be judged as well. Secondly, the major motivation for unpaid reviewers is often to build up their social capital in their networks of colleagues, so that responding diligently to review requests from leading academics in their field may lead to appointments to editorial boards, associate editors etc. This factor may be less present in many mega-journals, which rely more on the sense of academic duty of potential reviewers.

Another aspect limiting growth is the lack of branding for authors publishing in mega-journals. Many experts, promotion committees etc. base their evaluations of junior academics primarily on where these have published. In most fields there are well-established pecking orders of top journals; in addition, having a reasonably good impact factor also counts. Academics also tend to spread their articles over a range of journals where one or even two mega-journal articles can be acceptable, but where numerous ones in the list of publications would be not be good sign.

## ACKNOWLEDGEMENTS

Tea Mäkelä helped gather publication volume data from journal web sites, and her help is greatly appreciated.

### Funding

This work has been conducted as part of my professorial position at my University with no external funding. The funders had no role in study design, data collection and analysis, decision to publish, or preparation of the manuscript.

### Competing Interests

The author declares there are no competing interests.

### Author Contributions

- Bo-Christer Björk conceived and designed the experiments, performed the experiments, analyzed the data, contributed reagents/materials/analysis tools, wrote the paper, prepared figures and/or tables, reviewed drafts of the paper.

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
