# Peer review of "Have the "mega-journals" reached the limits to growth?"

_PeerJ, doi:10.7717/peerj.981_

## Round 0.1 · original submission · Major Revisions

All reviewers commented on the lack of a clear and consistent definition of a 'mega-journal' and I suggest you focus on this. I also agree with Reviewer #3 that the Introduction could be shortened. I do encourage you to revise the manuscript and resubmit as your study is interesting, original and topical, but it needs more clarity in the reporting.

Reviewer 1 ·

Basic reporting

The author often expresses opinions and asserts them as facts without supporting references. For me, the article needs to begin with a more robust definition of a megajournal, quantified as part of the first supported assertion in the introduction regarding the number of OA journals listed in the DOAJ. The author admits to the risk of missing out key journals (eLife?) from his research in line 142-3. The article and the premise of the research and subsequent conclusions rest on the definition and quantification data. Many of the assertions made in the rest of the introduction are ambiguous at best. e.g.Lines 29-32 re what the 'vast majority of academics seem to accept'; Lines 52-3 APCs are not charged for 'dissemination purposes'; Line 61 re 'most OA journals adhere to traditional peer review practices'. The article does not meet the criteria for basic reporting.

Experimental design

Flawed due to the issues referred to above regarding definition and identification of megajournals

Validity of the findings

As above

Additional comments

Perhaps start again...

·

Basic reporting

Background is lacking references for many claims stated.
Additionally, discussion should cite more papers on the topic.
The article should be structured according to authorship guidelines, with appropriate sections.
Tabe 2 is missing Journal and Year designations.

Experimental design

Table 2 – As many journals did not reach even 300 publications, why are they still considered Megajournals in your study?
An additional Table for the 13 journals in this study should contain the checklist for all the criteria of a mega-journal they have according to the authors, or at least an explicit statement that all shown in the table have the criteria chosen.

Validity of the findings

Although this is a descriptive study, results should be described with a much higher level of precision, based on the data collected, and in a format of - X out of 13 megajournals studied had this or that (i.e. - line 262 Why most? Didn’t you check for all the ones listed in table 1 and 2? Also, as you checked for Altmetrics info, and present it in Table 3, please describe the results in the same format x out of y.

Additional comments

First paragraph – some citation for this info would be welcomed.
Line 29 – The vast majority? Can you cite surveys for this, otherwise suggest toning it down.
Lines 67, 69 – citations please
Line 134 – which competitors?
Line 210 – could you not extract number of authors for all articles in the first quarter of 2015 for PeerJ and make an estimate of the cost ?
Furthermore I suggest the authors to look at STM report 2015 for number of articles per journals, as well as costs of publishing - http://www.stm-assoc.org/2015_02_20_STM_Report_2015.pdf
Line 228 – Strange wording, as megajournals also use 2-3 reviewers.
Lines 248-249 – citations please
Lines 273-278 – Have any surveys on this been done?
References should be corrected - line 366 doi?
line 369 ? belongs to the previous?
line 370 - add Available at as in the first reference
Lines 372 and 379 - have brackets for years, rest don't
Discussion - please start with the conclusion of your results - In this study we identified 13 MJ, whose....
Last paragraph of discussion - a summary of the discussion section is not needed, recommend deleting it.

·

Basic reporting

This paper is interesting and deserves publication after improvement. This topic deserves to be well explained to researchers. There are few good articles on the subject. The manuscript must be better structured (IMRaD?), especially the methods must be explicit because there are often comments added in the methods and results.

Introduction
1. The introduction is too long for an original article. Nevertheless its content is useful for the novice. I suggest the author to propose a box with this general information, and so an introduction would better clarify the purpose of research and highlight 'aim of this study' (see lines 123-132).
2. Line 66: do you mention 100 ‘original’ articles per year, or all types of articles?
3. The features mentioned from line 92 to 106 the criteria for a megajournal: were they used for the search of this study? They must appear in the methods’ section?
4. Line 117: add that “few journals (eLife) refuse the impact factor game”.

Experimental design

Methods are not precise
5. there must be a clear and precise definition of the criteria for inclusion in this study, and therefore better define the megajournal
6. given the subject matter, and the experience of the author, we accept vague phrases like in 'lines 140 to 142’. I was surprised to see the Table 1 (results) in the' methods' section. In this table 1, does not appear Frontiers, but appear journals that have a low volume of publication. How to understand that journals publishing fewer than 200 articles per year (Table 2) were defined as megajournals? All this would be avoided if the inclusion criteria were better defined. The research period should be mentioned.
7. In methods, appears a list of journals not included (line 161 Hindawi, etc ...) with comments to be in the discussion.
8. Why 3 journals from table 1 disappeared in table 2 (Royal Society Open Science, Brill Open Humanities, Heliyon? Probably because they did not publish papers…
9. Table 1: replace 'coming soon', by ‘expected June 2015’ if this is the case

Validity of the findings

Results
10. The border between results and methods is not clearly individualized. Most tables relate the results.
11. Table 2: the 2015 calculation should be updated on further versions of the manuscript
12. Acceptance rates are interesting, even if few journals provided data. Comments appear in this results section (line 220 to 228). They should be moved to the discussion part.
13. Publication speed: lines 230 to 232 are methods. There are references mentioned in this results section..
14. The part ‘interactive and updating features’ (line 251 to 265) is a comment, not a result.
15. Table 3 must have footnotes explaining the headings such as ‘multiple formats data’; the column ‘soundness review’ seems useless, if it’s the definition of megajournals.

Discussion
16. You must better discuss the methods, the definition of megajournals and their limits
17. I don’t know if predatory publishers created megajournals: is it a danger? The Jeffrey Beall has no information http://scholarlyoa.com/2014/10/28/shabby-indian-management-megajournal-reveals-its-peer-review-process/#more-4361
18. I did not find a clear answer of the question of the title.

References
They are fine, and some recent papers can contribute to the discussion on the peer-view process (is it still peer-review?). See “Mega-journals and peer review: can quality and standards survive?” http://pubs.acs.org/doi/pdf/10.1021/acs.chemmater.5b01142

Additional comments

this manuscript needs restructuring, probably under an IMRaD format, with better presentation of Methods and Results.

Minor comment: Should you write PLOS all over the paper and not PLoS

---

## Round 0.2 · Minor Revisions

Although both reviewers feel your article is improved, one reviewer would like more substantive change so I am giving you the option to revise once more. I also agree that the Abstract should be changed since the observation about the 35,000 articles/year relates to only one journal so it cannot be stated as a general conclusion. However, if your observation relates to the total pool of articles (which, presumably will be shared between the mega-journals) then this statement should be rephrased. I also agree with the reviewer that the term 'vote' is probably inappropriate. It is perhaps better to rephrase this, eg 'leaves it up to readers to decide what is of interest and importance to them' -- I think you may be confusing the selection process with post-publication features and Altmetrics.

·

Basic reporting

I find this article to seriously lack the standard IMRaD structure, with too much overlap between methods, results and discussion, and recommend complete rewriting.
Please fully report the methods in the methods section, for all the criteria used, and how you handled them. Leave the results section out of methodology explanation s and comparisons with other studies. Do the later in the discussion.
All tables lack names for columns and rows, table 4 should be redone, with one row for criteria.

Experimental design

Recommend removing Brill Open Humanities and Heliyon for the results, and mentioning them in the discussion, as they have not yet published any papers.
Furthermore I highly recommend the author to try and compare these 16 journals one to another, and suggest the practices they should learn one from another, or form other journals. I feel this is lacking in this study.
As in my previous commentary plz describe averages and ranges for all journals in results section for IF, APCs, volume numbers, publication times.
Additionally per Wikipedia article, it would be perhaps wise to mention self declared mega journals.

Validity of the findings

It is very unclear still what are the findings and what personal commentary.
Please make it clear also in the text, that only 6 of these journals have article numbers that can be considered higher than average for scientific journals (or more if you feel that to be true).

Additional comments

Abstract
vote with downloads and citations for the contribution? – What do you mean? Vote? IF is still calculated for many, as well as citation counts. Perhaps rephrase with greater focus on – altmetrics
However, alemetrics are now being used even for subscription based journals.

ebbed out at around 35,000 -- only one journal (PLOS) managed this, and sentence implies more did, so please be specific
No need for space before %
Other common features? Advise rephrasing as no common features are mentioned in the previous sentence.

Introduction
Line 41 – recommend loosing the footnote and using reference instead
Line 48 – can you start in the order of the paragraph before – first with numberless publiting, then a paragraph about APCs, and lastly peer review.
Line 64 few journals have exceeded 1,000 research articles per year – citation please, or list some examples that are not megajournals (e.g. xxx) -

Lines 75to 78 – recommend deleting
Line 92-95 should be transferred to discussion
Methods-
Recommended listing the aim in as the last paragraph of the introduction as a sentence instead of bulletin points.
I feel the methods section is too long, and should be thoroughly revised, just to shortly explain the criteria listed in the table 1. Furthermore, Wikipedia criteria – in the table should be reconsidered – as Wikipedia lists articles that have listed these, and those should be considered primary sources of information instead of Wikipedia.
Author should also acknowledge if he was or not a contributor to Wikipedia for mega journal entry.
Additionally, many sentences from results should be in this section.

Results
Recommend removing Brill Open Humanities and Heliyon for the results, and mentioning them in the discussion, as they have not yet published any papers. Lines 174 and 183 should not be in results but in methods (or as an appendix).
Also recommend the paragraphs in results to follow the order of criteria in table 1.
Line 188 should be left for discussion, and many similar sentences in results section. Plz report only results from journals in table 1 in the result section, and restrain for comparisons and comments.
Line 208-217 should be in methods
Line 226-228 in methods...

Discussion
Do not mention tables in discussion, use the numbers you want to report.
First paragraph should be restructured, and comparisons from results section mentioned here, to relate to your findings, otherwise it seems your findings are only 1 sentence, and all else commentary.
As in my previews review, delete the summary of the last paragraph, and find a better way to end your article. Abstract is used for summary.

·

Basic reporting

This new version is improved. It's easy to read.
There are still some comments in the methods section, but I do not have constructive remarks for this part. Anyway, the methods to exclude journals are not stated in the methods section : there are some infos in the results (lines 175, and 181/182).

I forgot to mention Collabra in my first review : why Collabra is not mentioned ? Is Collabra similar to Helyion ?

Lines 216 and line 220 : duplication of the info for the 2015 counts of articles.

The paper is really improved. The discussion should better explain why journals publishing less than 200 or 500 papers per year are included among the megajournals (see table 3). I am still surprised that small journals appear when Frontiers publishes 10 000 papers (is Frontiers very different from PLOS ONE?)

Experimental design

see above

Validity of the findings

see above

Additional comments

see above

---

## Round 0.3 · accepted · Accept

Thanks for your detailed response to the 2nd round of comments. (For your info, the 3rd reviewer declined to review again, and PeerJ requires only 2 reviewers' opinions.) I am now happy to accept your article.